# Susceptibility of Melanoma Cells to Targeted Therapy Correlates with Protection by Blood Neutrophils

**DOI:** 10.3390/cancers16091767

**Published:** 2024-05-02

**Authors:** Simone Wendlinger, Jonas Wohlfarth, Claudia Siedel, Sophia Kreft, Teresa Kilian, Sarah Junker, Luisa Schmid, Tobias Sinnberg, Ulrich Dischinger, Markus V. Heppt, Kilian Wistuba-Hamprecht, Friedegund Meier, Luise Erpenbeck, Elsa Neubert, Matthias Goebeler, Anja Gesierich, David Schrama, Corinna Kosnopfel, Bastian Schilling

**Affiliations:** 1Department of Dermatology, University Hospital Würzburg, 97080 Würzburg, Germany; 2Mildred Scheel Early Career Center Wuerzburg, University Hospital Wuerzburg, 97080 Würzburg, Germany; 3Department of Medical Oncology, The Christie NHS Foundation Trust, Manchester M20 4BX, UK; 4Division of Dermatooncology, Department of Dermatology, University of Tübingen, 72076 Tübingen, Germany; 5Department of Dermatology, Venereology and Allergology, Charité-Universitätsmedizin Berlin, 10117 Berlin, Germany; 6Department of Endocrinology and Diabetology, University Hospital Würzburg, 97080 Würzburg, Germany; 7Department of Dermatology, Universitätsklinikum Erlangen, Friedrich-Alexander-Universität Erlangen-Nürnberg, 91054 Erlangen, Germany; 8Skin Cancer Unit, German Cancer Research Center (DKFZ), 69120 Heidelberg, Germany; 9Department of Dermatology, Venereology and Allergology, University Medical Center Mannheim, Ruprecht-Karl University of Heidelberg, 68167 Mannheim, Germany; 10Department of Dermatology, Faculty of Medicine, University Hospital Carl Gustav Carus, Technische Universität Dresden, 01307 Dresden, Germany; 11Skin Cancer Center at the University Cancer Centre Dresden and National Center for Tumor Diseases, 01307 Dresden, Germany; 12Department of Dermatology, University of Münster, 48149 Münster, Germany; 13Leiden Academic Centre for Drug Research, Leiden University, 2333 Leiden, The Netherlands; 14Department of Dermatology, Venereology and Allergology, University Medical Center, Göttingen University, 37075 Göttingen, Germany; 15Department of Hematology, Oncology and Pneumology, University Hospital Münster, 48149 Münster, Germany

**Keywords:** melanoma, neutrophils, MAPK inhibition, resistance

## Abstract

**Simple Summary:**

Melanoma patients with high neutrophil counts often show impaired clinical response and poor prognosis, indicating that neutrophils can support melanoma progression. The precise mechanism responsible for this correlation, especially in the context of targeted therapy, still requires clarification. We show that peripheral blood neutrophils of patients with advanced melanoma are characterized by lower CD16 surface expression compared to healthy donors, which has been reported to be associated with tumor promotion. We provide evidence that melanoma cells under dual-targeted therapy can be protected in vitro by neutrophils from both patients and healthy donors. In addition, this protective effect is dependent on cell–cell contact, as well as on culture conditions, and is observed under nonadherence. Unraveling the mechanism, the interference with the protease activity of neutrophils reduced protection. Understanding the complex interaction of neutrophils and melanoma cells might aid in discovering methods to prevent the tumor-promoting effects by neutrophils in patients.

**Abstract:**

Elevated levels of peripheral blood and tumor tissue neutrophils are associated with poorer clinical response and therapy resistance in melanoma. The underlying mechanism and the role of neutrophils in targeted therapy is still not fully understood. Serum samples of patients with advanced melanoma were collected and neutrophil-associated serum markers were measured and correlated with response to targeted therapy. Blood neutrophils from healthy donors and patients with advanced melanoma were isolated, and their phenotypes, as well as their in vitro functions, were compared. In vitro functional tests were conducted through nonadherent cocultures with melanoma cells. Protection of melanoma cell lines by neutrophils was assessed under MAPK inhibition. Blood neutrophils from advanced melanoma patients exhibited lower CD16 expression compared to healthy donors. In vitro, both healthy-donor- and patient-derived neutrophils prevented melanoma cell apoptosis upon dual MAPK inhibition. The effect depended on cell–cell contact and melanoma cell susceptibility to treatment. Interference with protease activity of neutrophils prevented melanoma cell protection during treatment in cocultures. The negative correlation between neutrophils and melanoma outcomes seems to be linked to a protumoral function of neutrophils. In vitro, neutrophils exert a direct protective effect on melanoma cells during dual MAPK inhibition. This study further hints at a crucial role of neutrophil-related protease activity in protection.

## 1. Introduction

Over the last years, neutrophils have emerged from a restricted perception of them as innate immune cells merely participating in first-line host defense and are now allocated to the immune cell arsenal regulating tissue regeneration, affecting autoimmune diseases, diabetes and even cancer [1]. Neutrophils are defined by their functional plasticity during inflammatory processes and display flexible and complex bidirectional properties against cancer [2,3]. Neutrophils, the predominant leukocyte subset found in the peripheral bloodstream, exhibit variable counts influenced by their activation status [4]. Because of a complex concert of cytokines and activation states, neutrophils show both suppressing and promoting roles in different cancers [5]. In fact, high peripheral blood neutrophil counts have been linked to cancer progression. Additionally, a high neutrophil-to-lymphocyte ratio (NLR) has been described as a negative prognostic factor in patients with cancers, including advanced colorectal cancer, breast cancer and melanoma [3,6]. Furthermore, a study by Szczerba et al. underlined the intriguing effect of blood-derived neutrophils, showing a proliferation-favoring and metastasis-promoting interaction of neutrophils with circulating tumor cells (CTCs) [7]. The formation of neutrophil–cancer cell clusters highlighted the direct effect of neutrophils on tumor progression in melanoma and was linked to a worse progression-free survival in patients with breast cancer [7,8,9].

Regulated by a cocktail of chemokines and chemoattractants, such as CXCL6 and CXCL8, peripheral blood neutrophils are recruited into the tissue (e.g., lung or liver) and, as subjects of a cancerous disease, have been shown to shape the tumor microenvironment (TME) [2]. Aside from blood neutrophils, clinical studies also highlight the unfavorable association of tumor-associated neutrophils (TANs) with clinical response in patients with breast cancer and melanoma [10,11,12,13,14,15,16]. This effect is highly dependent on the type of cancer and the complex interplay with the adaptive immune system. Interestingly, a study conducted by Slattery and colleagues revealed the necessity of a functional ICAM-1/Mac-1 adhesion axis accompanied by a cocktail of cytokines responsible for the migration of melanoma cells and favoring the aggregation of melanoma cells to neutrophils under flow conditions [17]. Thus, neutrophils appear to directly and indirectly promote tumor progression and facilitate early tumor dissemination. This is supported, for example, by the production and release of elastase and MMP9 or factors inducing angiogenesis, such as VEGF, as well as through adhesion-enhancing tumor migration and protection of CTCs in the blood [17,18,19,20,21,22]. Likewise, melanoma cells can activate neutrophils demonstrating a bidirectional interaction of both neutrophils and melanoma cells in favor of the latter [23,24].

Aside from their phagocytic capacity, an arsenal of granules packed with antimicrobial proteases and peptides such as myeloperoxidase (MPO), neutrophil elastase (NE) and the formation of NETs are properties attributed to neutrophils [25]. NETs are DNA structures expelled by neutrophils during inflammation and upon exposure to stimuli such as bacterially derived lipopolysaccharide (LPS) [25,26]. NET formation is regulated by enzymes, including the protein arginine deiminase (PAD) 4, NE, and MPO, in the initiation phase followed by the rupture of the neutrophil nuclear membrane due to high pressure caused by chromatin decondensation [26]. During the plasma membrane rupture, the DNA-scaffold associated with granule products like proteases is released into the extracellular space. Growing evidence accentuates the protumor effect of NETs in cancer [25,26]. Indeed, NETs promote metastasis by initiation of proliferation of dormant breast cancer cells in mice and in an in vitro model in a protease-dependent manner [26]. Just recently, a significant positive correlation of NET amounts and tumor size was found in a melanoma metastases cohort, underlining the clinical risk attributed to neutrophils [27]. Importantly, cancer therapy including immunotherapy and targeted therapy also possess immune-modulating features shaping the tumor microenvironment [28,29,30]. For instance, BRAF and MEK inhibitors potentiate T-cell infiltration and intensify the antitumor T-cell capacity [29]. Nonetheless, how treatment influences neutrophil polarization and function is complex and not yet well understood [31]. However, potential mechanisms are described, which may favor tumor resistance to therapy. A study conducted by Mousset and colleagues reported the formation of NETs during chemotherapy induced by cytokine release of cancer cells. The NET formation caused TGF-β activation, which induced a switch in cancer cells from an epithelial-to-mesenchymal phenotype, a state associated with resistance to BRAF inhibitors and chemotherapy [32,33,34].

This study analyzes the impact of peripheral blood neutrophils in patients with advanced metastatic melanoma receiving first-line targeted therapy. Hence, correlation of neutrophil-associated serum markers with clinical outcome and phenotypical, as well as functional, examinations of patient-derived neutrophils were conducted. To gain a more comprehensive understanding of the impact of neutrophils on malignant melanoma, we conducted in vitro experiments characterizing the functional interaction between melanoma cells and neutrophils, among others, comparing healthy-donor- and patient-derived neutrophils. These experiments involved coculturing melanoma cells and neutrophils under conditions mimicking both adherent and nonadherent environments, as found in the blood and in tissue. We specifically explored the effects of dual-targeted therapy with BRAF/MEK inhibitors on these cocultures. Viability assays and cell cycle analysis were utilized as primary readouts. Additionally, we explored intervention in neutrophil-associated protease activity and NET function in nonadherent cocultures.

## 2. Materials and Methods

### 2.1. Patient Cohort and Healthy Donors

A cohort of eight patients with stage IV melanoma prior to treatment implementation, three patients with stages I and II melanoma with no evidence of disease (NED) and seven healthy donors (HD) as controls were included for ex vivo phenotypical analysis of peripheral blood neutrophils. Treatment-naïve patients with newly diagnosed stage I or II melanoma were enrolled after surgical treatment. For the in vitro functional comparison, an additional thirty-two patients with stage IV melanoma prior to treatment administration and ten healthy donors were included. A different healthy donor was used for each replicate for the in vitro functional analyses, including all nonadherent, adherent and 3D cultures. Further clinical parameters were not collected for this cohort (Table 1) [35].

For a second cohort (Table 2), the patient inclusion criteria were defined and 145 serum samples were included as described in Wendlinger et al. [35]. All patients from the second cohort receiving either targeted therapy or immunotherapy prior to and during treatment with no restriction to a certain therapy line were enrolled, with the exclusion of adjuvant therapy (listed in Table 2), and were used for serum analysis. In short, patients with newly diagnosed metastatic or unresectable cutaneous melanoma presenting at the Department of Dermatology, University Hospital Würzburg, were enrolled between October 2012 and November 2019 [35]. The ethics committee of the University of Würzburg (50/17-mk) approved the study, and all patients enrolled in this study provided written informed consent in accordance with the Declaration of Helsinki. A total of 53 of the 145 retrospective serum samples from advanced melanoma patients receiving immunotherapy as a first-line therapy were received from the Department of Endocrinology and Diabetology at the University Hospital Würzburg. We kindly received 16 retrospective serum samples from advanced melanoma patients receiving targeted therapy as a first-line therapy from the Department of Dermatology, University Hospital Erlangen, and 7 serum samples from the multicentric blood bank of the Department of Dermatology, University Hospital Tübingen, which were derived from patients recruited at the Department of Dermatology, University Hospital Dresden (Table 2) [35]. Pretreatment and on-treatment serum samples were collected and correlated with the clinical response defined as responders (CR = complete remission or PR = partial response) and nonresponders (PD = progressive disease) to the respective treatment according to RECIST 1.1 [36]. In patients undergoing dual-targeted therapy, the median duration between the initial sample (pretreatment) and the subsequent sample (on-treatment) was 102 days, with a range of 70–139 days. On-treatment serum samples were collected in proximity to the initial response assessment. For patients receiving immunotherapy, the median duration between the pre- and on-treatment samples was 171 days, with a range of 46–286 days. The white blood cell counts and serum lactate dehydrogenase (LDH) values were captured up to 63 days before the collection of both the pretreatment and on-treatment serum samples. When multiple values were accessible, the nearest peripheral blood draw was taken. Demographic and clinical data were collected for all patients listed in Table 2 (an adapted list from Wendlinger and colleagues) [35].

### 2.2. Isolation of Neutrophils

Neutrophils were isolated from heparinized peripheral blood samples obtained from healthy volunteers and patients as previously described [35]. To separate neutrophils from eosinophils, an automatic magnetic labeling-based system, autoMACS pro, was used with a multi-antibody eosinophil isolation kit from Miltenyi Biotech (Bergisch Gladbach, Germany). Neutrophils were purified by positive selection, as previously described [35].

### 2.3. Comparing Neutrophil Isolation Methods

As a second neutrophil isolation method and for comparison of neutrophil function, neutrophils were isolated using a dual density–separation method with a combination of Histopaque^®^-1119 (Sigma-Aldrich, Darmstadt, Germany) and a second gradient density separation using Percoll^TM^ PLUS (GE Healthcare Bio-Sciences AB, Uppsala, Sweden), as described before [25].

### 2.4. Purity and Phenotyping of Neutrophils

The purity of isolated neutrophils and their phenotypic characterization was evaluated by flow cytometry, as described previously [35]. Neutrophils were identified as CD45^+^/CD16^+^/CD66b^+^/CD193^−^. A high purity of ≥90% was routinely obtained. For phenotypic characterization, 5 × 10^5^ neutrophils were stained with the following surface antibodies to define the neutrophil population: anti-CD16-FITC or -PB, anti-CD193 (CCR3)-PE or -APC-Cy7 (both Biolegend, San Diego, CA, USA) and anti-CD66b-APC (both eBioscience, Carlsbad, CA, USA). Additional to the lineage staining, one fluorochrome-labeled antibody for the following target antigens were used: CD69, CD31, CD29 (all Biolegend, San Diego, CA, USA). Measurements were performed using a CytoFLEX LX flow cytometer (Beckman Coulter, Krefeld, Germany).

### 2.5. ELISA

To determine galectin-3 levels in the sera of patients with advanced melanoma (*n* = 20), at the time of blood draw the sample was centrifuged at 800× *g* for 10 min, aliquoted and stored at −80 °C until use [35]. Galectin-3 was measured by ELISA (Sigma Aldrich #RAB0661, Darmstadt, Germany) using the manufacturer’s protocol and recommendations. Patient serum was diluted two-fold using the sample diluent buffer provided. For each sample, a duplicate was measured. Galectin-3 levels were correlated with the response of the respective patient to treatment with either targeted therapy or immunotherapy. Absorbance was measured at 450 nm using the Infinite M Nano plate reader (Tecan, Männedorf, Switzerland). As galectin-3 was measured after the assessment of MPO, MMP-9, HGF and IL-8 (see Section 2.6) in serum, a reduced number of samples were available.

### 2.6. Multiplex Analysis

A LegendPlex^TM^ (Biolegend, San Diego, CA, USA) multi-analyte analysis was used with a customized human panel to measure MPO, MMP-9, HGF and IL-8. A total of 50 pre- and 32 on-treatment serum samples from patients with advanced melanoma receiving targeted therapy were processed according to the manufacturer’s protocol. Patients receiving immunotherapy were used as a second cohort (*n* = 92). Concentrations (displayed as pg/mL) of the analytes in the serum were correlated with clinical response to treatment [35]. Milliplex^®^ human immune-oncology checkpoint protein panel 2 (Millipore #HCKP2-11K, Billerica, MA, USA) was used to measure galectin-1 and galectin-3 in 14 pre- and 14 on-treatment sera of patients receiving immunotherapy. Targets were measured according to the manufacturer’s protocol.

### 2.7. Cell Lines

The cell lines used in this study are listed in Table 3. All melanoma cell lines carry the BRAF^V600E^ mutation. The human melanoma cell lines MaMel51, MaMel63a, MaMel86c and MaMel06 were derived from patient biopsies as described previously [37]. The MaMel51 cell line was used as a standard model for the functional experiments and culture behavior in the context of neutrophil-induced protection. Additional BRAF-mutated melanoma cell lines (451LU, WM3734, Sk-Mel-5, M14, Sk-Mel-28, UACC257 and MDA-MB-435) were used as controls to corroborate the observed effects (Table 3). The cell lines 451LU and WM3734 were kindly provided by Dr. rer. nat. Heike Niessner from the Division of Dermatooncology, Department of Dermatology, University of Tübingen. Nonmelanoma-derived and non-BRAF-mutated cell lines, like the non-small-cell lung cancer cell line H460, the lung adenocarcinoma cell line A549 and the Merkel cell carcinoma cell line WaGa, were used as controls to evaluate the specificity of the neutrophil-mediated protection of cancer cells depending on their susceptibility to targeted inhibition including inhibitors specific for mutated BRAF. Cells were grown at 37 °C with 5% CO_2_ in RPMI-1640 medium (Sigma-Aldrich, Darmstadt, Germany) containing 10% FCS (Sigma-Aldrich, Darmstadt, Germany) and 1% penicillin/streptomycin (Sigma-Aldrich, Darmstadt, Germany), referred to as complete medium (CM), as previously described [35]. Cell lines were used within two months after thawing, and possible infections with mycoplasma were routinely checked for in all cell lines using MycoSPY Master Mix (Biontex, Munich, Germany).

### 2.8. Protection Assays

CFSE-labeled (2 µM, ThermoFisher Scientific, Eugene, OR, USA) melanoma or nonmelanoma cancer cells were cocultured with freshly isolated peripheral blood neutrophils at a target-to-neutrophil (T:N) ratio of 1:10. For all in vitro coculture experiments, cells were kept in CM or in CM containing 1 µM vemurafenib (PLX4032, Cayman Chemical, Denver, CO, USA) and/or 0.1 µM cobimetinib (GDC-0973, Cayman Chemical, Denver, CO, USA) (VC) or in CM supplemented with 20 µM cisplatin (CIS; kindly provided by the pharmaceutical department at the University Hospital Würzburg) as indicated. Transwell experiments were performed as described previously [35]. (Co-)Culture experiments were performed under nonadherent culture conditions in polypropylene tubes (Beckman Coulter, Brea, CA, USA), under adherent conditions in 24-well flat-bottom plates (Greiner Bio-One, Kremsmünster, Austria) or under 3D growth conditions (Section 2.13) for 24 h. For experiments with conditioned medium, the supernatant of melanoma cell–neutrophil cocultures after 24 h was harvested. Fresh CFSE-labeled MaMel51 cells were nonadherently cultured with conditioned medium or fresh CM or VC as controls, for 24 h. The viability of the melanoma cells or neutrophils was determined by 7-amino-actinomycin (7-AAD, ThermoFisher Scientific, Eugene, OR, USA) and Annexin V-APC (BD Biosciences, Franklin Lakes, NJ, USA) staining, as previously described [35]. Viable cells were defined as 7-AAD- and Annexin V-double negative. Measurements were performed using a CytoFLEX LX flow cytometer (Beckman Coulter, Krefeld, Germany).

### 2.9. Cell Cycle Analysis

A combination of 2’-deoxy-5-ethynyluridine (EdU, Biosynth, Berkshire, UK) and Hoechst 33,342 (Sigma-Aldrich, Darmstadt, Germany) staining was used to analyze the cell cycle distribution of nonadherently cocultured CFSE-labeled melanoma cells with neutrophils and treatment with VC after 24 h, as described previously [38].

### 2.10. Lysis of Neutrophils and Inactivation of Contents

To expose the intracellular neutrophil granule content, neutrophils were prepared as previously described [35,39]. For inactivation of exposed neutrophil content, samples were incubated at 95 °C for one hour utilizing a block heater (Stuart, Stone, UK). Lysed or heat-inactivated neutrophils were resuspended in CM and cultured with CFSE-stained melanoma cells for the indicated time intervals [35].

### 2.11. Time-Dependent Protection of Melanoma Cells

To determine the optimal timing for adding neutrophils to exert their positive impact on the viability of melanoma cells treated with VC, CFSE-stained MaMel51 cells were cultured in polypropylene tubes in CM or VC for 24 h with or without freshly isolated neutrophils. After 24 h, either fresh neutrophils were added or control solution (CM without neutrophils) was added for another 24 h. After a total of 48 h, the viability of the MaMel51 cells was determined as described above.

### 2.12. Cytospins and HE Staining

Nonadherent and adherent 24 h melanoma cell–neutrophil cocultures were processed as described above in Section 2.8. Cytospins and hematoxylin and eosin (HE) stainings were performed as described previously [35]. Imaging was carried out using a TI-E microscope (Nikon, Düsseldorf, Germany).

### 2.13. 3D Spheroid Assay

Melanoma cells were cultured using a classic hanging drop culture method to generate 3D spheroids, as previously described [40,41]. Drops were applied to the lid of a Petri dish filled with sterile H_2_O. At day seven, spheroids were imaged using the Leica DM750 microscope, and the medium was exchanged with either fresh CM or VC. After 48 h of treatment, images were taken and freshly isolated neutrophils were added for another 24 h. After a total of 72 h, the spheroid sizes were imaged. For cocultures, representative spheroids were gently washed to remove neutrophils and placed on a fresh Petri dish for imaging. The sizes of the spheroids were quantified using Fiji (Image J, Version 1.53c), as previously described [41]. For the viability assessment, spheroids in the same condition were pooled, washed with PBS, and stained with anti-CD16-PB for 30 min at RT in the dark to differentiate between neutrophils (CD16^+^) and melanoma cells. Subsequently, cells were washed with PBS and the viability was assessed as described above.

### 2.14. Target Blocking Experiment

Blocking potential target structures for the interaction of melanoma cells and neutrophils was conducted using 2 µg/mL anti-CD11a, anti-CD11b, anti-CD18, anti-CD54 (ICAM-1) or all blocking antibodies together in nonadherent MaMel51-neutrophil cocultures, as described in Section 2.8, for 24 h. An isotype control antibody (IgG1) served as a control. All antibodies were purchased from Biolegend (San Diego, CA, USA) and fulfilled the low endotoxin and azide-free (LEAF) specification.

### 2.15. Visualization and Interference with NETs

Nonadherently cultured neutrophils or cocultures with melanoma cells with or without treatment with dual BRAF/MEK inhibitors were transferred onto pretreated (99% alcohol) glass cover slips. The staining procedure was carried out as described previously [25,42]. Cells were stained with anti-human myeloperoxidase (MPO) (IgG, mouse, 1:500) (ab25989, Abcam, Cambridge, UK) and anti-human citrullinated histone H3 (H3cit) (IgG, rabbit, 1:750) (ab5103, Abcam, Cambridge, UK). For visualization, secondary fluorochrome-labeled antibodies (polyclonal anti-mouse Alexa555 (IgG, goat, 1:2000) (A-21422, ThermoFisher Scientific, Eugene, OR, USA), polyclonal anti-rabbit Alexa488 (IgG, goat, 1:500) (A-11034, ThermoFisher Scientific, Eugene, OR, USA)) were used. Chromatin was stained with Hoechst 33,342 (Sigma-Aldrich, Darmstadt, Germany). Cells were imaged with a fluorescence microscope (AxioImager M1, Software: AxioVision Rel.4.7, Zeiss, Macquarie Park, NSW, Australia). For dissolving the DNA scaffold of NETs, we treated melanoma cell–neutrophil nonadherent cocultures exposed to dual BRAF/MEK inhibition with DNase I (concentration 5 µg/mL; Roche, Mannheim, Germany) for 24 h. Protease activity was blocked treating cocultures for 24 h either with the cOmplete^TM^, Mini, EDTA-free protease inhibitor cocktail from Roche (stock concentration: 25× in H_2_O, 1× final concentration in CM; Roche, Mannheim, Germany) or with the EDTA-free protease inhibitor cocktail from Selleckchem (stock concentration: 100× in DMSO, 1× final concentration in CM; Houston, TX, USA). Appropriate controls with H_2_O or DMSO with the same final concentration were carried out for either protease inhibitor cocktails.

### 2.16. Statistical Analysis

The comparisons of the neutrophil phenotypes from patients and healthy donors were analyzed using Analysis of Variance (ANOVA) with Bonferroni correction for normally distributed data and the Kruskal–Wallis test with Dunn’s correction for non-normally distributed data. The normality of the data was assessed using the D’Agostino–Pearson omnibus test. An unpaired *t*-test was applied for comparison of the serum markers for responders and nonresponders. In vitro experiments on neutrophil–melanoma cell interaction were analyzed using ANOVA with Bonferroni correction for normally distributed data for three or more unmatched groups. Linear regression was used for the group correlations. Unpaired *t*-tests were used for two-group comparisons. Prism (Graph-Pad, version 7) was utilized for data visualization and statistical analysis. ns, *p* > 0.05, * *p* ≤ 0.05, ** *p* ≤ 0.01, *** *p* ≤ 0.001 and **** *p* ≤ 0.0001.

## 3. Results

### 3.1. Low MPO and HGF Serum Levels Are Associated with Response to Targeted Therapy and Immunotherapy in Patients with Advanced Melanoma

Since elevated peripheral blood neutrophils have been linked to poorer clinical outcomes across different cancers, such as melanoma, we evaluated a possible correlation between serum proteins associated with neutrophil function and response to MAPK inhibitors [5,43,44,45,46]. In this regard, we measured galectin-3, MPO, matrix metalloproteinase 9 (MMP-9), hepatocyte growth factor (HGF) and interleukin 8 (IL-8) in sera of 50 advanced melanoma patients (stage IV) receiving MAPK inhibitors as first-line therapy, and evaluated their relevance as predictive serum biomarkers comparing responders (PR) and nonresponders (PD). We could show that pretherapeutic galectin-3, MMP-9, HGF and IL-8 levels were similar in both responders and nonresponders to targeted therapy (Appendix A, Figure 1A). However, a higher pretherapeutic MPO serum concentration was associated with a better response to treatment, which did not reach statistical significance (*p* = 0.18) (Figure 1A). This observation was reversed for on-treatment MPO values, with responders showing a significantly lower MPO sera concentration (Figure 1B). Additionally, responders were characterized by significantly lower on-treatment HGF and by trending lower IL-8 (*p* = 0.05) values (Figure 1B). Galectin-3 can be secreted by both tumor cells and immune cells and high galectin-3 expression mediates sensitivity to vemurafenib in melanoma cells [47]. Although we could not find a correlation of circulating galectin-3 levels with response to targeted therapy, we wondered whether there are correlations between serum galectin-3 and the absolute leukocyte, absolute neutrophil, and relative neutrophil counts of patients before treatment with first-line MAPK inhibitors. Correlation tests were performed separately for responders and nonresponders (Appendix A). No significant association was found for absolute leukocyte count and galectin-3. However, a high pretreatment galectin-3 serum concentration positively correlated with high absolute and relative neutrophil counts in responders (*p* = 0.12, r^2^ = 0.25 and *p* = 0.11, r^2^ = 0.22 respectively), while for nonresponders negative correlations were observed for absolute and relative neutrophils counts (*p* = 0.25, r^2^ = 0.24 and *p* = 0.10, r^2^ = 0.43 respectively), showing, however, no statistical significance. Thus, we generally hypothesize that for responders showing high RNCs, the accompanied high galectin-3 levels in the serum might counteract the disadvantageous impact of neutrophils on the response to targeted therapy as described in the literature [10,47].

Importantly, the value of biomarkers and their impact on treatment response might vary depending on the specific treatment regimens. To this end, we also analyzed the sera from patients receiving immunotherapy. In line with the literature, high galectin-1 values were associated with nonresponse to immunotherapy treatment, showing, however, no statistical significance (*p* = 0.16) (Appendix A). In addition, nonresponders were characterized by high pretherapeutic HGF (*p* = 0.11) serum values. During drug administration, the responders to immunotherapy showed significantly lower serum concentrations of galectin-3 and galectin-1 compared to the nonresponders (Appendix A). The on-treatment MMP-9 values were significantly lower in the responders. Interestingly, the responders showed significantly lower serum concentrations of MPO, HGF and IL-8 for both immunotherapy and targeted therapy, which might hint at the independence of the prognostic value of these markers from the specific treatment.

### 3.2. Peripheral Neutrophils from Stage IV Melanoma Patients Prior to Drug Administration Show Lower CD16 Expression Compared to Healthy Donors

Next, we investigated whether the negative prognostic association of high peripheral blood neutrophils with the clinical outcome aligns with a certain phenotype of neutrophils in melanoma patients. We analyzed six previously described neutrophil surface markers and compared pretreatment neutrophils from stage IV melanoma patients to neutrophils from healthy donors. Stages I and II melanoma patients served as a control for patients with no current evidence of disease (cohort listed in Table 1). Subsequently, surface molecules including activation markers (CD66b, CD69), immunoregulatory receptors (CD16, CCR3) and adhesion molecules (CD31, CD29) were investigated [48,49,50]. Flow cytometry dot plots are shown as representative gating strategy in Appendix A. We could show that stage-IV-melanoma-derived neutrophils (patient IV) displayed significant lower CD16 expression compared to healthy-donor-derived neutrophils (HD) and a reduced expression compared to neutrophils from stages I and II melanoma patients (Patient I/II), which did not reach statistical significance (*p* = 0.27) (Appendix A). This suggests a correlation between CD16 expression on neutrophils and clinical status in patients with advanced melanoma compared to healthy individuals. Additionally, a tendency toward lower CD31 (*p* = 0.09) and CCR3 expression (*p* = 0.10), and higher CD69 (*p* = 0.07) and CD29 expression (*p* = 0.15) was observed on neutrophils from late-stage melanoma patients compared to healthy donors, which were all not statistically significant. Stages I and II melanoma-derived neutrophils showed significantly higher CD66b expression than neutrophils obtained from healthy donors and by trend a higher expression compared to neutrophils from late-stage melanoma patients (*p* = 0.09) (Appendix A).

### 3.3. Peripheral Blood Neutrophils from Patients and Healthy Donors Similarly Prevent BRAF-/MEK-Inhibition-Induced Apoptosis of Melanoma Cells In Vitro

As many clinical and in vitro studies reveal tumor-promoting effects exerted by peripheral blood neutrophils in melanoma, we hypothesized a direct protective effect mediated by neutrophils on melanoma cells [51]. We contemplated the existence of a potential difference in functionality of neutrophils comparing healthy donors and patients. Firstly, we evaluated the impact of healthy donor neutrophils on the viability of melanoma cells after BRAF/MEK inhibition using in vitro cocultures. The MaMel51 cell line, harboring a BRAF^V600E^-mutation, provided a suitable model for 24 h cocultures to investigate neutrophil-mediated protection. To this end, we analyzed the viability of MaMel51 cells under nonadherent culture conditions exposed to BRAF/MEK inhibition for 24 and 48 h in the absence or presence of neutrophils. Nonadherent cultures were used to mimic the environment found in the peripheral circulatory system. A ratio of 1:10 tumor cell (T) to neutrophils (N) was used. A representative gating strategy is shown for assessing viability of melanoma cells to visualize the effect by neutrophils in cocultures (Appendix A). Neutrophils reversed the cytotoxic effect of vemurafenib or cobimetinib single treatment and of combinatory treatment in MaMel51 cells after 24 h culture under detached conditions, shown as an increase in MaMel51 cell viability compared to samples without neutrophils (control) (Figure 2A left). After 48 h, the positive effect of neutrophils toward MaMel51 cells was reduced and only a trend toward improved melanoma cell viability was observed (Figure 2A right). Neutrophils are highly sensitive immune cells with varying function depending on their polarity and stimulation [2,5]. To exclude an effect on the activation and functionality of neutrophils by the utilized isolation technique, we compared two neutrophil isolation methods (Appendix A). Both separation techniques yielded a neutrophil fraction with a similar mean viability of >80% immediately after isolation (Appendix A). Importantly, using either method we could reproduce a similar rescue of nonadherently cultured MaMel51 cells by neutrophils in treated conditions with BRAF/MEK inhibition after 24 h (Appendix A). Consequently, for further analyses the Biocoll separation followed by a magnetic separation was used (Appendix A). To verify the observed effect by neutrophils toward melanoma cells, we further analyzed eleven BRAF-mutated melanoma cell lines and three nonmelanoma and non-BRAF-mutated cancer cell lines under nonadherent culture conditions in the absence or presence of neutrophils exposed to BRAF/MEK inhibition or cisplatin for 24 h. We disclosed varying degrees of protection for the tested cell lines toward exposure to BRAF/MEK inhibition and to cisplatin as a BRAF-independent treatment control after 24 h by freshly isolated peripheral blood neutrophils (Figure 2B, Appendix A, Table 4). Protection was defined as a positive effect on melanoma cell viability by at least 10% increase when cultured with neutrophils, while an effect below 10% was defined as unprotectable. The best-protected melanoma cell lines were 451LU, followed by MaMel51 cells, with an increase in viability for BRAF-/MEK-inhibitor-treated samples of 42% and 35% respectively compared to cultures without neutrophils (Figure 2B left, Table 4). In cisplatin-treated conditions, 451LU cells settled as the third best, and MaMel51 cells as the seventh best-protected cell line with a positive effect on viability by neutrophils of 28% and 18% respectively (Figure 2B right, Table 4). This hints at a general positive effect of neutrophils toward stressed cancer cells independent of the respective treatment. In comparison, the MDA-MB-435 cell line persisted as the least affected cell line upon BRAF/MEK inhibition or cisplatin treatment and, thus, could not show any protection by neutrophils. Nonmelanoma cell lines including the lung carcinoma cell line H460, the lung adenocarcinoma cell line A549 and the Merkel cell carcinoma cell line WaGa displayed reduced cell viability upon coculture with neutrophils when treated with BRAF/MEK inhibitors highlighting the dependence of effector to target cell when exploring neutrophil function (Figure 2B; Table 4). In addition, we observed a positive correlation between the susceptibility of melanoma cells to BRAF/MEK inhibition, as well as cisplatin treatment, and their protection by neutrophils (Figure 2C). We therefore concluded that impairment of viability upon treatment was required for the visualization of protection mediated by neutrophils in in vitro nonadherent cultures after 24 h.

For the evaluation of the neutrophil-induced protection of melanoma cell viability upon treatment, we next compared the function of freshly isolated neutrophils from healthy donors to pretreatment stage IV melanoma patients (cohort listed in Table 1) in cocultures with MaMel51 cells after 24 h. Additionally, cells were exposed to dual BRAF/MEK inhibitors. Neutrophils derived from both healthy donors and patients exhibit comparable favorable effects on melanoma cells when cultured in medium and when exposed to treatment after 24 h (Figure 2D). Remarkably, the degree of protection highly depended on the neutrophil donor. Additionally, an increase in the melanoma cell viability was significantly more prominent in the treated cocultures (Figure 2D).

### 3.4. Melanoma Cells and Neutrophils Engage in a Reciprocal Relationship In Vitro

To explore whether melanoma cells and neutrophils share a beneficial bidirectional relationship, we analyzed the neutrophil viability in nonadherent cocultures with melanoma cell lines. As therapy might influence neutrophil function and surface marker expression, we additionally exposed the cocultures to BRAF-/MEK-inhibitor treatment and as a control to cisplatin for 24 h. After 24 h in culture, neutrophil viability was significantly diminished (mean viability < 30%) compared to neutrophil viability immediately post-isolation (mean viability > 80%) (Appendix A). However, neutrophil viability remained unperturbed by either BRAF/MEK inhibitors or cisplatin after 24 h. Interestingly, cocultures with cell lines including MaMel51, 451LU, WM3734 and Sk-Mel-5, which showed strong impairment of viability upon BRAF/MEK inhibition, displayed a positive effect toward neutrophil viability. This effect was observed in the culture medium with BRAF/MEK inhibitors and when treated with cisplatin, as shown by the increase in the neutrophil viability; however, this was not statistically significant (Appendix A). The superiority in the neutrophil viability could not be reproduced when cultured with cell lines that showed less or no impairment in viability upon BRAF/MEK inhibition, including MaMel63a, Sk-Mel-28, UACC257 or MDA-MB-435 cells (Appendix A). We concluded that melanoma cells enhance the neutrophil viability after 24 h in nonadherent cultures, which reflects a bidirectional dependence of the cell populations in regard to the longevity in vitro. In addition, the lifespan prolongation seemed to depend on the melanoma cell line and was more prominent in all conditions when cocultured with melanoma cells that showed pronounced impairment of viability upon treatment in the viability assays, such as MaMel51 cells.

### 3.5. Neutrophils Protect Melanoma Cells from BRAF-/MEK-Inhibition-Induced Cell Cycle Arrest

Since BRAF/MEK inhibitors induce both apoptosis, as well as a cell cycle arrest, in melanoma cells, we used the cell cycle analysis as an alternative readout to clarify the unresponsiveness of cell lines such as Sk-Mel-28, UACC257 and MDA-MB-435 in the viability readouts to treatment under nonadherent conditions after 24 h [52,53]. We exposed Sk-Mel-28 and UACC257 cells to dual BRAF/MEK inhibition for 24 h. The 451LU and MaMel51 cells were analyzed as control cell lines, which already showed impaired viability upon treatment in the viability assays. We revealed a significant G1-arrest and a reduction in the S-phase in all four tested cell lines upon treatment for 24 h (Appendix A). Thus, we conclude that all of the tested BRAF-mutated melanoma cell lines are susceptible to treatment but the response differs and might be predominantly visible in increased apoptosis and/or cell cycle arrest. To understand the protective effect by neutrophils in this context, cell cycle analyses of cocultures were also conducted for MaMel51 and UACC257 cells with neutrophils in the absence or presence of treatment. Neutrophils significantly resolved the G1-arrest in UACC257 cells induced by the treatment and even induced a significant reduction in the G1-phase in medium (Appendix A). This event was accompanied by an induction of the S-phase by neutrophils in treated UACC257 cells, however, it did not show statistical significance (*p* = 0.19). A trend toward a reduced S-phase for treated MaMel51 cells was observed. The S-phase of treated MaMel51 cells increased in coculture with neutrophils; however, it did not reach statistical significance. Considering the observations from both the viability assay and the cell cycle analysis, the results suggest that neutrophils possess melanoma-cell-promoting abilities independent of the respective treatment option. However, the effects depend on the tested cell line and the cell lines’ mode of response to dual BRAF-/MEK-inhibitor treatment. Additionally, the treatment-induced cytotoxic effect might also depend on the duration of the treatment exposure.

Strikingly, examining the cell line origin, their apoptotic rate after the BRAF/MEK inhibition and protection by neutrophils in viability assays, four-sevenths of the cell lines with impaired cell viability upon BRAF/MEK inhibition were lymph-node-derived, one cell line (Sk-Mel-5) originated from a brain metastasis and one (MaMel06) from a lung metastasis (Table 3). In comparison, three-quarter of the cell lines with less impaired cell viability upon BRAF/MEK inhibition originated from a skin metastasis, and one cell line (UACC257) could not be assigned to the site of origin. We hypothesize that even when analyzing the same cancer entity, the site of origin of the cell line might further hint at their susceptibility to apoptosis or cell cycle changes upon treatment.

### 3.6. Cell–Cell Contact Dictates the Active Process of Neutrophil-Mediated Protection of Melanoma in the Context of Dual BRAF/MEK Inhibition

To understand the relationship between melanoma cells and neutrophils, we next wanted to narrow down the functional framework in which the protective effect by neutrophils is evident. To this end, we performed transwell experiments to verify the role of cellular contact and soluble mediators. Neutrophils and MaMel51 cells were either kept separated by a semipermeable membrane or combined as cocultures in the bottom of a transwell tube for 24 h (Figure 3A). Importantly, the separate coculture abrogated the positive effect of neutrophils toward melanoma cell viability (Figure 3B left). This was not the case in the combined coculture, in which an increased melanoma viability was detected when directly cocultured with neutrophils in treated conditions, suggesting a mechanism requiring adjacency of the effector to target cell (Figure 3B right). The need of proximity was confirmed in experiments with conditioned medium showing a lack of protection for melanoma cells when exposed to dual BRAF/MEK inhibition compared to control cocultures (Figure 3C). As neutrophils contain a plethora of granules and granular content, we wondered whether the release of such neutrophil products would interfere or enhance protection. MaMel51 cells were cocultured with fresh, lysed or heat-inactivated lysed neutrophils in indicated medium for 24 h. Lysis prevented neutrophils from exerting their protective function toward MaMel51 cells against dual BRAF/MEK inhibition and cisplatin and even showed direct cytotoxicity toward MaMel51 cells (Figure 3D). Heat inactivation reversed the direct neutrophil-mediated killing, but the neutrophils remained unable to increase the melanoma cell viability upon exposure to both BRAF/MEK inhibition and cisplatin, rendering protection an active process requiring intact neutrophils with unperturbed cellular content. Interestingly, when narrowing down the time point at which neutrophils needed to be added to the coculture to rescue melanoma cells, we can show that the neutrophils introduced to melanoma cells, either immediately or 24 h later, alongside targeted therapeutics, exhibited a comparable protective effect (Figure 3E). The sustained impact of melanoma cell protection against BRAF and MEK inhibition notably increased when freshly isolated neutrophils were added to the coculture at both time points, thus creating resistance by melanoma cells toward targeted therapy even for a longer period of at least 48 h.

### 3.7. Neutrophil-Induced Protection of Melanoma Cells Is Observed in Nonadherent and 3D Cultures

Initial cocultures were performed under nonadherent conditions mimicking detached properties and forcing direct the interaction of effector and target cells. Next, we explored the effect of adherent and 3D cultures on the investigated effect in melanoma provided by neutrophils (Figure 4A). Adherent cultures were used to mimic the interaction observed in tumor tissue. Adherent and pooled (adherent and detached cells) MaMel51 cells showed by trend (*p* = 0.07 and *p* = 0.15, respectively) an impairment in viability when exposed to dual BRAF/MEK inhibition (Figure 4B). However, the impact of the treatment was not sufficient to induce significant protection by neutrophils in terms of melanoma cell viability. The viability of the detached melanoma cells in adherent cultures was, however, significantly and positively influenced by the presence of neutrophils, thereby restoring the protective effect against dual BRAF/MEK inhibition. Cytospins of cells in the supernatant and subsequent HE staining of adherent and detached cells (without BRAF/MEK inhibition) revealed the formation of cell–cell aggregations between MaMel51 cells and neutrophils in the supernatant, while the adherent cells showed a lack of cell–cell proximity, potentially explaining the absence of protection in the adherent setup (Figure 4C). Switching to a 3D culture system with melanoma cell lines capable of forming spheroids, both the 451LU and MDA-MB-435 melanoma cell lines displayed dampened and even reduced tumor spheroid sizes (area µm^2^) upon treatment with dual BRAF/MEK inhibitors for a total of 72 h (Figure 4D,E). Adding neutrophils to 3D melanoma cell spheroids did not affect tumor sizes in cultures with medium and only by trend (*p* = 0.06) reduced the spheroid sizes of 451LU but not MDA-MB-435 cells exposed to treatment. Analyzing the viability of the (co-)cultured spheroids revealed a significant viability reduction upon BRAF/MEK inhibition for both tested cell lines (Figure 4F). Importantly, a significant increase in viability was observed for both 451LU and MDA-MB-435 cells when adding neutrophils to MAPK-inhibitor-treated conditions. We therefore hypothesized that neutrophils protect melanoma cell viability from treatment under nonadherent cell culture conditions including 3D culture models, an environment more likely to be encountered in the blood stream. This tumor-favoring effect by neutrophils might, ultimately, facilitate metastatic spreading of melanoma cells in the peripheral blood.

### 3.8. Neutrophil Extracellular Traps form in Melanoma Cell–Neutrophil Cocultures and Protease Inhibitors Prevent Protection of Melanoma Cells by Neutrophils

Our data hint at a cell–cell contact-dependent mechanism provided by neutrophils toward melanoma cells. To unravel the underlying mechanism, we tested surface adhesion markers expressed on melanoma cells and neutrophils, including CD11a, CD11b, CD18 and IACM-1, as potential function-mediating targets [54]. Since the monoclonal blocking antibodies directed against these specific adhesion markers in the MaMel51–neutrophil cocultures did not show any functional impacts; we assume that another receptor or ligand is probably required for mediating protection by neutrophils (Appendix A).

The release of the granule contents, such as proteases, and the formation of NETs has been linked to the progression of melanoma and even resistance to therapy [32,55]. The exact mechanism is still under investigation. We confirmed the appearance of NETs in melanoma cell–neutrophil cocultures by fluorescence staining of markers associated with NETs, including chromatin, MPO and citrullinated histone 3 (H3Cit), and performed experiments interfering with neutrophil-associated protease activity and NETs (Figure 5A–D). The latter was accomplished by either dissolving the DNA web-like structures using DNase I in cocultures or interfering with the protease activity using two kinds of protease inhibitor mixtures (Figure 5C,D). DNase I treatment was unable to prevent protection under dual BRAF-/MEK-inhibitor treatment (Figure 5C). Using the cOmplete protease inhibitor mix from Roche successfully and significantly prevented neutrophil-induced protection of the MaMel51 cells against dual BRAF/MEK inhibition compared to the coculture without protease inhibitor and coculture with the respective inhibitor control (here, H_2_O) (Figure 5D). A second protease inhibitor mix purchased from Selleckchem caused a reduced effect of neutrophils; however, it did not reach statistical significance. A trend toward increased MaMel51 viability was observed when cocultured with neutrophils and treatment compared to the culture without neutrophils (*p* = 0.14). Aside from cellular proximity, we can show the relevance of intact neutrophil-associated protease activity for the melanoma-cell-promoting effect in the context of BRAF/MEK inhibition in vitro.

## 4. Discussion

Various studies provide insight into the potential function of neutrophils in melanoma and their role in resistance mechanisms to cancer therapy [2,3]. Tumor-promoting properties of neutrophils have been shown in patients with melanoma, as high peripheral blood neutrophils correlate with inferior survival and cancer progression [3,6,10,51]. However, whether the clinical perception of neutrophils as a bad prognostic marker in melanoma is reflected by serum markers associated with neutrophils and their function is poorly understood. Several studies have shown that elevated levels of MPO, MMP-9 and IL-8 are associated with poor prognosis and compromised immune response in melanoma [45,56,57]. We could show that while advanced-melanoma patients responding to targeted therapy were characterized by higher pretreatment MPO levels, the pretreatment MMP-9, HGF and IL-8 levels were similar in both responders and nonresponders. It is important to note that differences between responders and nonresponders to drug initiation could be caused by various factors, including patient biology, treatment pharmacokinetics, tumor characteristics, and individual treatment responses [58,59]. In fact, galectin-3 expression is generally reduced in progressing melanoma cells, and in the context of targeted therapy, it was reported to induce sensitivity of melanoma cells to treatment [60]. In our hands, for first-line targeted therapy patients, the pre- and on-treatment galectin-3 levels were similar between responders and nonresponders. Importantly, the significance of biomarkers and their influence on treatment response may exhibit variability depending on the particular treatment. In contrast to the targeted therapy, high galectin-3 expression was rather associated with worse clinical outcome, and pretherapeutic circulating galectin-3 and galectin-1 predicted poor treatment response for immunotherapy with PD-1 blockade [61,62]. In the context of immunotherapy, we could show that responders displayed lower galectin-1 concentrations in their sera, which is in accordance with the literature [61,62]. Unlike galectin, the MPO, HGF and IL-8 levels seemed rather like treatment-independent prognostic factors, as responders for both targeted therapy, as well as immunotherapy, were characterized by significantly lower serum levels on-treatment. Evaluating a selection of previously described neutrophil-associated serum biomarkers, our analysis revealed comparable pretreatment serum concentrations among both responders and nonresponders for both targeted therapy and immunotherapy. Consequently, we assume that the investigated markers may not be suitable for the prediction of treatment response for either therapy approach. A reassessment of these markers using a more extensive patient cohort may offer elucidation.

Aside from neutrophil-associated serum markers, we assumed that the presented neutrophil phenotype might differ between patients and healthy donors and potentially offer functional insight into the disadvantageous association of neutrophils in melanoma. CD16 (Fc gamma RIII) has been demonstrated to play an important role in antibody-dependent cellular cytotoxicity (ADCC), contributing to the effective elimination of primary cancers and cancer cell lines [63]. In fact, the existence of infiltrating myeloid cells expressing high levels of CD16 is correlated with enhanced survival outcomes in colorectal cancer patients and other cancers, while CD16^low^ neutrophils have been reported to correlate with treatment resistance in patients [16,48]. CD16, together with CD11b and CXCR4, is associated with neutrophil maturation [50]. We show that neutrophils from patients with advanced melanoma prior to drug administration displayed a significant decrease in surface CD16 expression and by trend a CD69^high^CD29^high^CD31^low^CCR3^low^ phenotype compared to neutrophils from healthy donors. Because of the small cohort, we could not analyze the association of CD16 expression with clinical response to treatment. Thus, whether CD16 expression also predicts the response to treatment in the context of melanoma remains to be clarified. This warrants further, more detailed investigations in the framework of future studies with a higher number of patient samples. Moreover, our study did not encompass a broader patient cohort, particularly regarding patients with early-stage melanoma. Consequently, the examination of differences between early-stage and late-stage melanoma patients in comparison to healthy controls was limited, thus precluding comprehensive conclusions. On the basis of the preceding clinical data and considering the diminished CD16 expression observed ex vivo in patient neutrophils, we assume that the disease status of a patient might define the prevalence of a certain neutrophil population in melanoma.

Investigating the unfavorable link between peripheral blood neutrophils, in the context of melanoma, and targeted therapy, we firstly examined impairment of viability upon targeted therapy in eleven BRAF-mutated melanoma cell lines and observed varying degrees of sensitivity to treatment. Despite harboring the BRAF mutation as the common denominator of the tested melanoma cell lines, melanoma cells displayed varying inter-tumoral heterogeneity affecting treatment response and potential resistance [64,65]. The BRAF-mutated melanoma cells exhibited varied responses to treatment, including decreased viability and/or induction of a G1 arrest in the cell cycle analysis. This is in line with different modes of response of molecularly heterogeneous melanoma cell lines to MAPK-pathway inhibitors [52,53]. Importantly, cocultures with neutrophils induced a strong prolongation of melanoma cell viability under treated nonadherent and two-dimensional culture conditions. Intriguingly, in parallel to a significantly increased fraction of viable cells, we also noted a seemingly reduced size of melanoma spheroids after coculture with neutrophils and concomitant MAPK-inhibitor treatment. This reduced spheroid size could be caused by decreased adhesive properties of the melanoma cells following coculture with neutrophils, as has also been observed in the framework of adherent two-dimensional coculture conditions. Detached melanoma cells might still benefit from neutrophils as proposed by a higher number of viable cells in the supernatant of adherent cultures. Therefore, we hypothesize that the observed increased proportion of viable melanoma cells in cocultures might not only be a consequence of a protection of the melanoma cells by neutrophils but might also be facilitated by an additional selective pressure favoring resistant and viable melanoma in vitro, which requires further investigations. In addition, the degree of protection highly depended on the neutrophil donor. The favorable impact of neutrophils on melanoma cells was not confined to targeted therapy; it was also evident in cisplatin-treated conditions, thereby expanding the neutrophil-induced protection to other treatment regimens, which can be applied independent of the prevalence of BRAF mutations. Most importantly, we could also show that patient and healthy donor-derived neutrophils exhibit similar protective abilities toward melanoma cells treated with targeted therapy. This finding is surprising as one could assume different functions of peripheral blood neutrophils in patients compared to healthy donors due to several factors. One such factor might be the tumor microenvironment, which could alter neutrophil activation and chemotaxis through tumor-derived factors [66,67]. Additionally, patients tend to exhibit increased inflammation compared to healthy donors, which might affect neutrophil function [68]. However, as our study focuses on neutrophils of the peripheral blood, we cannot exclude an effect of inflammation or tumor-associated factors on tissue-associated neutrophils. Importantly, changes in surface characteristics might still exert considerable effects on neutrophil functionality in vivo. An impact of the changes in surface expression of the Fc gamma RIII CD16 on neutrophil function might significantly depend on the presence of critical factors including relevant antibodies and various immune cells facilitating ADCC. The approach used in our in vitro study lacks such factors, which precludes analysis of this aspect of neutrophil function and might explain the observed comparable neutrophil function from both patients and healthy donors. Thus, to achieve a more comprehensive understanding of neutrophils comparing patients and healthy donors in vitro, the tumor microenvironment and the arsenal of immune cells present in vivo should be considered when addressing neutrophil function in future investigations. In addition, we cannot exclude direct or indirect effects on patient-derived neutrophil function during treatment, since, in our study, neutrophils for phenotypic characterization and functional analyses were obtained from patients prior to the start of treatment. A more extensive evaluation of the neutrophil phenotype, also in respect to CD16 surface expression, and correlation with the activation status, as well as the specific function, might help develop a better understanding of neutrophils in melanoma.

Demonstrating a reciprocal influence on cell viability, we delineate a bidirectional relationship between melanoma cells and neutrophils. Neutrophils typically exhibit a brief lifespan, with a half-life in the circulation of approximately 8 h, although tissue-resident neutrophils may survive for up to two days [69,70,71]. Prolonged survival of tissue-associated neutrophils is correlating with a tumor-promoting phenotype [72]. Notably, a significant portion of neutrophils in in vitro cultures undergo apoptosis within 18 h [70]. We observed a trend indicating enhanced neutrophil viability in coculture with melanoma cell lines under untreated, BRAF-/MEK-inhibitor-treated and cisplatin-treated conditions. This is in line with literature showing prolonged neutrophil survival in cultures with melanoma stem cells and breast cancer cells [24,73]. It should be noted that varying extents of neutrophil viability have been reported in different in vitro culture studies, which might partially be caused by the use of different isolation or culture techniques [74]. Excluding a bias using a certain isolation technique, we compared two different isolation methods and obtained similar post-isolation neutrophil viability accompanied by similar neutrophil function in regard to melanoma cell protection.

Initial coculture experiments were performed under nonadherent conditions to mimic the situation in the circulatory system as encountered by circulating tumor cells. The polypropylene tubes used in this study may not precisely replicate the conditions found in the human peripheral blood system due to several limitations. Unlike the dynamic and complex environment in the circulatory system, polypropylene tubes lack continuous flow and shear stress. The latter might influence neutrophil–melanoma cell interaction and CTC clustering. However, we assumed that polypropylene tubes might create similar conditions due to lower binding properties compared to polystyrene tubes for adherent cultures. Using a combination of microfluidic systems including perfusion might optimize our method and achieve a further refined resemblance of blood conditions. While CTC clusters are infrequent in the circulatory system, they exhibit a high metastatic potential [75]. The occurrence of CTC–neutrophil clusters positively correlate with tumor metastasis [7,8,76]. Kiniwa et al. reported an occurrence of around 2.15 CTCs in 1 mL blood from melanoma patients, and we isolated around 1.2 × 10^6^ neutrophils per 1 mL blood, which would present a ratio of around 1:550,000 melanoma cell to neutrophils in the blood [77]. We decided to use a 1:10 melanoma-cell-to-neutrophil ratio, which appeared more manageable during the in vitro experiments and was already sufficient to induce a robust neutrophil-mediated effect on melanoma cells. We illustrated the requirement for direct cell–cell contact and emphasized the importance of the integrity of an intact cellular membrane for neutrophils to exhibit their protective effect. Furthermore, we found that introducing neutrophils to cocultures after 24 h of melanoma cell treatment enhances melanoma cell survival even at later culture time points. Thus, the immediate presence of neutrophils with melanoma cells exposed to therapeutics is not necessary for the long-term enhanced survival of melanoma cells. We showed that nonadherent and 3D but nonadherent cocultures with neutrophils enhanced melanoma cell viability upon treatment with dual-targeted therapy. Nonetheless, TANs are also associated with unfavorable clinical prognosis in several cancer entities and have a limited response to immunotherapy [12,13,14,15]. Thus, we cannot, per se, exclude a positive effect of neutrophils on melanoma cells under adherent conditions or in tissue in which the tumor microenvironment plays a role. The visualization of coculture interaction revealed melanoma cell–neutrophil aggregation, confirming and highlighting the importance of cellular proximity for the establishment of a protumoral neutrophil function. Examining CTCs in breast cancer patients and their connection with white blood cells, Szczerba et al. discovered the aggregation of cancer cells and neutrophils [7]. This aggregation was associated with an increase in cancer cell cycle progression, indicating that interactions with neutrophils confer a proliferative advantage to CTCs. Interestingly, we also showed an increase in the S-phase and a reduced G1-phase in treated UACC257 cells in nonadherent cocultures with neutrophils after 24 h, reflecting cell cycle progression. Thus, to improve the targeted therapy response and simultaneously reduce the neutrophil effects, it would be important to examine the mode of response of melanoma cells to treatment, reflected by the extent of the impaired viability or cell cycle.

CD11, CD18 and ICAM-1, which are relevant for neutrophil recruitment, migration and function, posed as potential targets to abrogate the observed protective effect. In vitro studies show a direct interaction of lung cancer cells and breast cancer cells with neutrophils facilitating tumor growth and metastasis into the vasculature [78,79]. However, testing blocking antibodies against adhesion receptors including CD11 and CD18, we could not prevent neutrophil-mediated protection in our settings. Interestingly, the same study revealed the involvement of neutrophil elastase in the pro-proliferative effect toward lung cancer cells [78]. Neutrophil elastase, a serine protease, and further neutrophil-related proteases can be either released by neutrophils through secretion, degranulation or by NETosis [80]. MMP-9 and MPO are components of NETs present on externalized nuclear DNA that promote endothelial cell and cancer cell proliferation [81,82]. Studies also hint at the ability of NETs to capture circulating tumor cells, escorting them through the circulatory system and even promoting adhesion and extravasation [83]. Exploring the relevance of NET formation in melanoma-cell–neutrophil cocultures, we can confirm the presence of NETs in nonadherent cocultures by the visualization of citrullinated histone H3 (H3cit), MPO and DNA (via Hoechst) both in untreated and treated conditions with dual-targeted therapy. Yet, the resolving of NET DNA scaffolds, as reported by Cools-Lartigue, using DNase I treatment showed no effects in our hands [83]. Importantly, we could prevent neutrophil-mediated protection toward a melanoma cell line in treated conditions by suppressing protease activity using a cocktail of protease inhibitors. In line with this finding, Albrengues and colleagues demonstrated metastasis promotion by NET-associated proteases in a lung model [26]. Protease activity was pivotal for this effect and remained unaffected by DNA scaffold dissolution. Consequently, preserved protease function in our cocultures after DNase I treatment could be assumed, potentially allowing for the unaltered protection of melanoma cells by neutrophils. The possible necessity of the DNA scaffold for protease activity in vivo, possibly facilitating a localized high protease concentration, cannot be excluded on the basis of our finding. In addition, whether protease release and activity are a result of granule secretion by neutrophils or NETosis remains to be clarified.

## 5. Conclusions

Investigating the disadvantageous prognostic effect of neutrophils in melanoma, our study shows that peripheral blood neutrophils in late-stage melanoma patients display reduced CD16 expression compared to healthy donors, a phenotype associated with treatment resistance and inferior survival, for example, in patients with colorectal cancer [48,49]. In vitro, the patient- and healthy-donor-derived neutrophils exhibited similar protective abilities toward dual BRAF-/MEK-inhibitor-treated melanoma cells after 24 h. Additionally, we demonstrated a strong correlation of melanoma cell susceptibility to dual BRAF/MEK inhibition and the degree of protection in cocultures with neutrophils. Importantly, we could prevent protection by interference with neutrophil-associated proteases, thus highlighting the potential importance of protease activity in this context. Further research on the exact mechanism driving the correlation of neutrophils with a worse clinical outcome in melanoma should provide insight into the complex interaction and might help in finding a way to reduce or fully prevent the tumor-promoting effects by neutrophils.

## Figures and Tables

**Figure 1 cancers-16-01767-f001:**
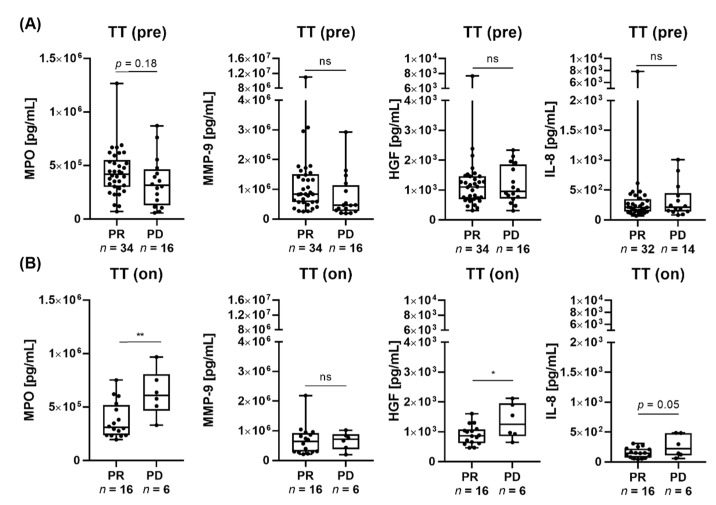
Exploring serum markers and their correlation with clinical outcomes in patients with advanced metastatic melanoma: (**A**,**B**) comparison of serum MPO, MMP-9, HGF and IL-8 concentrations (pg/mL) of responders (PR) and nonresponders (PD) (**A**) prior (TT (pre)) and (**B**) during (TT (on)) treatment with first-line targeted therapy. There is a trend toward lower pretherapeutic MPO levels in the nonresponders compared to the responders. This observation was significantly reversed for MPO levels during treatment. The sera of the responders presented lower HGF and IL-8 levels for on-treatment. In total, 50 patients with metastatic melanoma pretreatment (PR, *n* = 34; PD, *n* = 16) and 32 patients on-treatment (PR, *n* = 16; PD, *n* = 6) were included in the targeted therapy cohort. ns, *p* > 0.05, * *p* ≤ 0.05, ** *p* ≤ 0.01.

**Figure 2 cancers-16-01767-f002:**
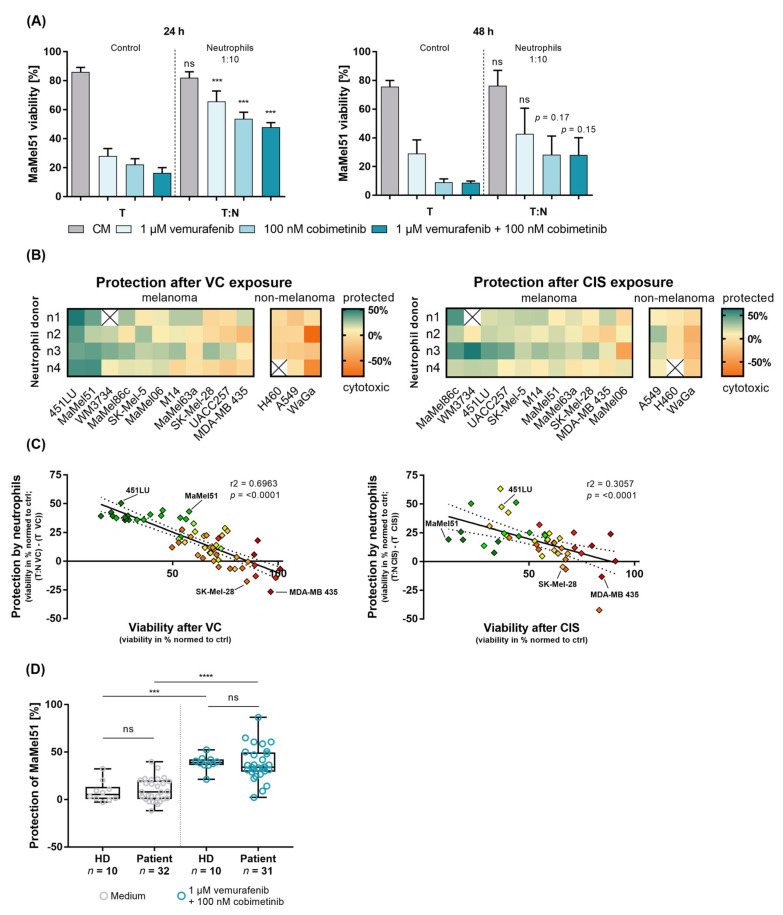
Both patient- and healthy-donor-derived neutrophils protect melanoma cells from apoptosis induced by MAPK inhibition in vitro. Nonadherent (co-)cultures of BRAF-V600-mutated melanoma cells and nonmelanoma cancer cells in the absence or presence of peripheral blood neutrophils from (**A**–**D**) healthy donors or (**D**) stage IV melanoma patients in complete medium (CM), CM containing 1 µM vemurafenib and/or 100 nM cobimetinib (VC) or 20 µM cisplatin (CIS) for 24 h if not indicated otherwise. (**A**–**C**) Neutrophil-induced protection of BRAF-mutated melanoma cell lines under treatment with dual BRAF/MEK inhibition or cisplatin. (**A**) (Co-)Culture of MaMel51 cells with or without neutrophils at a 1:10 ratio (T:N) for (**left**) 24 and (**right**) 48 h with indicated treatment. Neutrophils significantly increased MaMel51 cell viability when treated with vemurafenib or cobimetinib single treatment and with combination treatment (VC) after 24 h but not after 48 h. Significances compared to respective controls without neutrophils are shown. Mean percentage of MaMel51 cell viability + SD is shown for two to six independent experiments. (**B**) Protective effect of neutrophils toward melanoma and nonmelanoma cancer cell lines in conditions with (**left**) VC or (**right**) CIS after 24 h. The melanoma or nonmelanoma cell viability was normalized to untreated controls. The viability of cocultured-treated cells was subtracted from the viability of treated cells alone to calculate the percentage of protection (viability^T:N VC^-viability^TVC^ or viability^T:N CIS^-viability^TCIS^) and each value was visualized in the heatmap. Squares are colored by melanoma/nonmelanoma cell-based protectability. Increase in cell viability by 10% compared to control was considered as protected by neutrophils (green). Neutrophils exhibiting cytotoxicity toward treated cell lines was plotted as red. Protection by neutrophils depended on the tested cell line, with 451LU being the best protected and MDA-MB-435 being the least protected melanoma cell line in VC-treated conditions. Neutrophils were cytotoxic toward VC-treated nonmelanoma cancer cell lines. Data from three to four independent experiments are listed. (**C**) Correlation of VC and CIS protection by neutrophils and melanoma cell viability after (**left**) VC or (**right**) CIS treatment is shown for each melanoma cell line and each replicate. Susceptibility to either treatment strongly correlated with the protectability of the tested cell line. Each color represents one cell line. (**D**) Comparison of MaMel51 protection by neutrophils from healthy donors (HD) and stage IV melanoma patients (Patient) after 24 h. Nonadherent cocultures in complete medium or 1 µM vemurafenib and 100 nM cobimetinib. Viability of MaMel51 cells was assessed to address neutrophil effect and viability was normalized to the respective control. Protection was calculated by subtraction of normalized viability of cocultured untreated or treated cells from the normalized viability of untreated or treated cells alone (viability^T:N CM^-viability^TCM^, viability^T:N VC^-viability^TVC^ or viability^T:N CIS^-viability^TCIS^) and each value was visualized. The mean percentage of the MaMel51 cell protection ± SD is shown for independent experiment; *n* = 10 healthy donors, *n* = 31 to 32 patients pretreatment. ns, *p* > 0.05, *** *p* ≤ 0.001 and **** *p* ≤ 0.0001.

**Figure 3 cancers-16-01767-f003:**
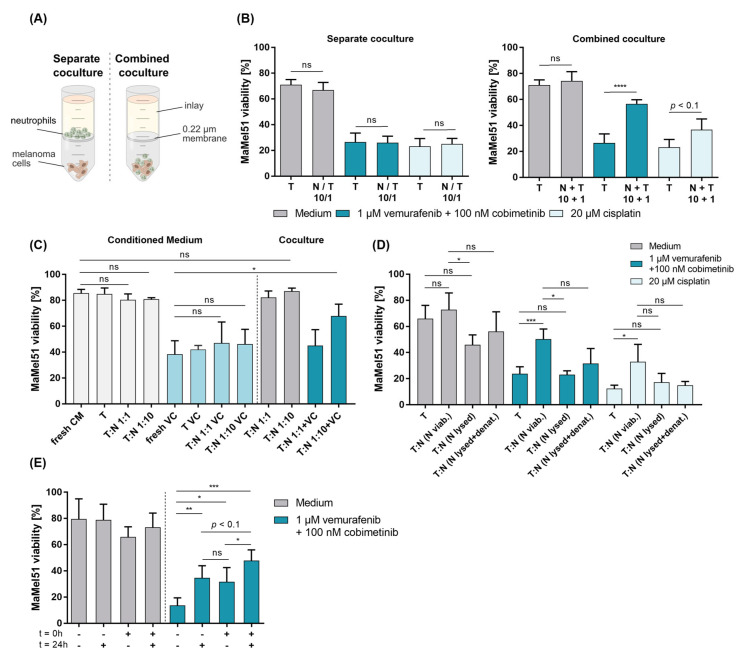
Physical proximity is crucial for the functionality of neutrophils in cocultures with melanoma cells. (**A**–**C**) Melanoma cell protection by neutrophils required cell–cell contact. (**A**) Graphical demonstration of separate (**left**) and combined (**right**) cocultures of MaMel51 cells with neutrophils at a ratio of 1:10 (T:N) in Spin-X^®^ columns containing a semipermeable membrane (pore size: 0.22 μm) for 24 h analyzed in (**B**). Culture with or without 1 μM vemurafenib and 100 nM cobimetinib (VC) or 20 μM cisplatin. (**C**) Nonadherent cultures of freshly prepared MaMel51 with conditioned medium (supernatant) from previous MaMel51 cocultures with neutrophils were carried out for 24 h. Fresh CM with or without the addition of 1 μM vemurafenib and 100 nM cobimetinib or 20 μM cisplatin was used as a control. Nonadherent cocultures with freshly isolated neutrophils were prepared as a further control. (**D**) Nonadherent cocultures of MaMel51 cells with or without viable, lysed or heat-inactivated neutrophils at a ratio of 1:10 in the indicated medium for 24 h. Neutrophils acquired cytotoxic abilities in medium once lysed and could not exert their protective function toward MaMel51 cells exposed to VC or CIS treatment. The heat inactivation of the neutrophil content was not sufficient to restore the protective function. (**E**) Time-dependent rescue of MaMel51 cells by neutrophils. MaMel51 cells were nonadherently cultured with (+) or without (−) the addition of neutrophils at a 1:10 (T:N) ratio in indicated medium for a total of 48 h. Protection in VC-conditions could be restored in MaMel51 cells when adding neutrophils after 24 h and was enhanced when neutrophils were added at both 0 h and 24 h time points. Mean percentage of MaMel51 cell viability + SD is displayed from (**B**) three to five, (**C**) three, (**D**) three to six and (**E**) four to five independent experiments. ns, *p* > 0.05, * *p* ≤ 0.05, ** *p* ≤ 0.01, *** *p* ≤ 0.001 and **** *p* ≤ 0.0001.

**Figure 4 cancers-16-01767-f004:**
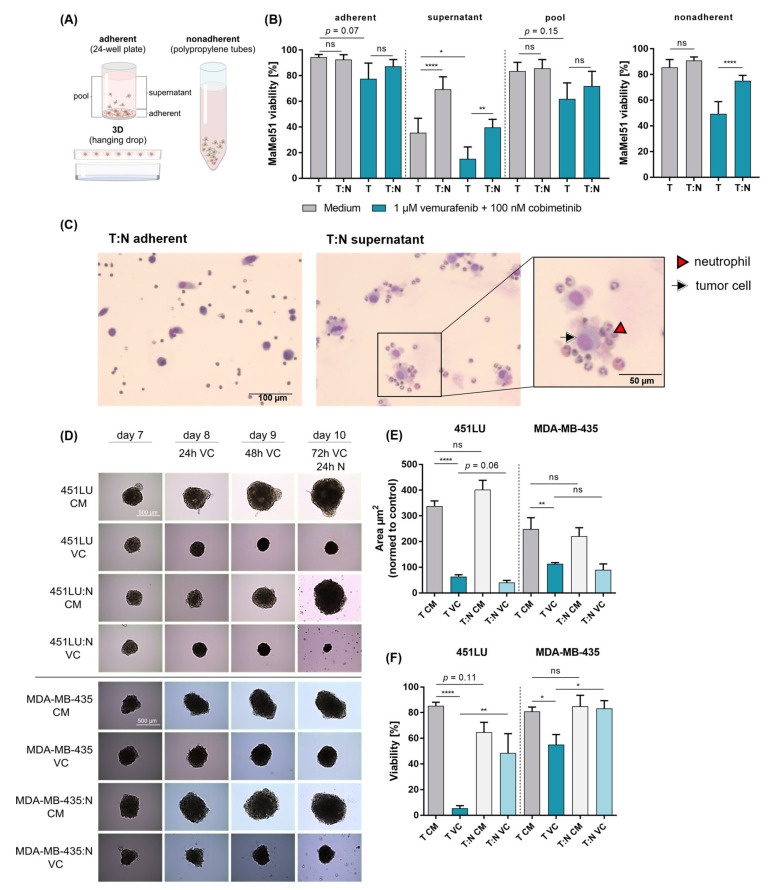
Neutrophil-induced protection against BRAF/MEK inhibition is restricted to detached cells and nonadherent cultures. (**A**) Graphical illustration of adherent and nonadherent cocultures in 24-well plates and polypropylene tubes, respectively. The pooled sample represents adherent cells and cells harvested from the supernatant. Three-dimensional cultures were performed using a hanging drop technique. (**B**) MaMel51 cell viability of adherent cells, detached or pooled cells after coculture with or without neutrophils in indicated medium for 24 h compared to nonadherent cocultures. The positive effect of neutrophils toward melanoma cell viability is only observed for detached MaMel51 cells of adherent cultures or nonadherent cultures when exposed to VC treatment. The effect was also observed for detached MaMel51 cells cultured in medium. Mean melanoma cell viability + SD is shown for three to five independent experiments. (**C**) Formation of neutrophil–melanoma cell aggregates in the supernatant of adherent cocultures (T:N supernatant). Aggregation was not seen for cells still attached to the culture vessel (T:N adherent). The red arrow points to a neutrophil. The black arrow indicates a melanoma cell. Scale bar = 100 µm. (**D**) Representative spheroids for 451LU and MDA-MB-435 cell growth as spheroids in medium or medium containing 1 µM vemurafenib and 100 nM cobimetinib (VC) with or without neutrophils for a total of 72 h (day 10). After 48 h of VC treatment or culture in CM as indicated, freshly isolated peripheral blood neutrophils were added for 24 h. Scale bar represents 500 µm. (**E**) Spheroid size for (**D**) each condition was quantified and normalized to spheroid size captured before treatment (day 7). Mean size (area in µm^2^) + SD is shown for two to three (each with n ≥ 3) independent experiments. (**F**) Viability assessment of 451LU and MDA-MB-435 spheroid cells captured in (**D**) after a total of 72 h of culture. Both 451LU and MDA-MB-435 cells show susceptibility toward VC treatment. The protective effect of neutrophils was visible for both tested cell lines in VC-treated conditions. The mean melanoma cell viability + SD is shown for two to three independent experiments. ns, *p* > 0.05, * *p* ≤ 0.05, ** *p* ≤ 0.01 and **** *p* ≤ 0.0001.

**Figure 5 cancers-16-01767-f005:**
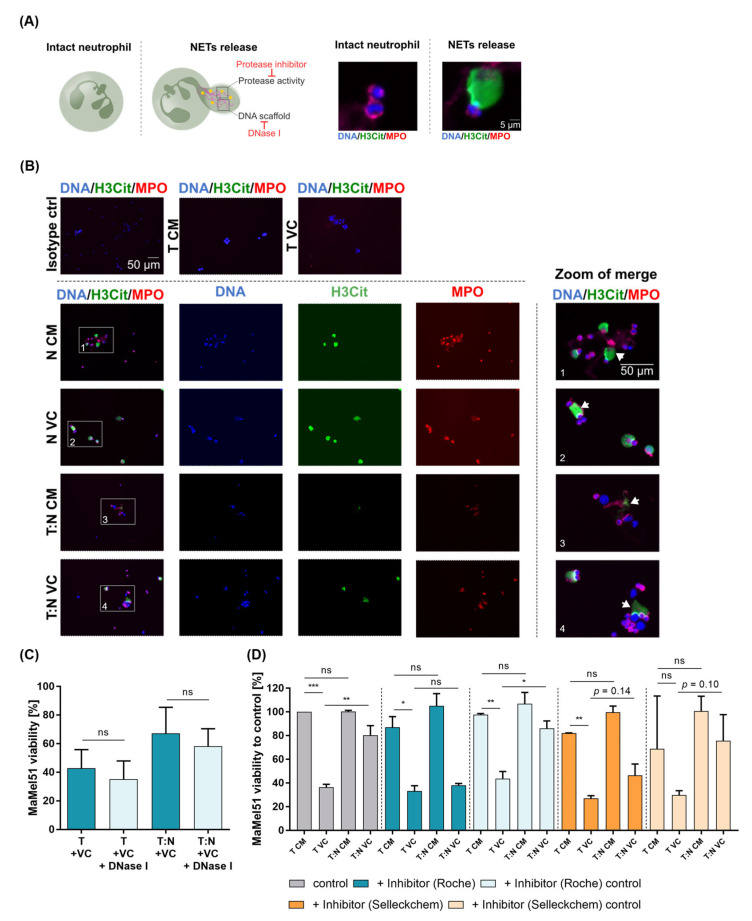
Interference with neutrophil-associated protease activity prevents protection in vitro. (**A**) Graphical illustration (**left**) of an intact neutrophil and a neutrophil releasing NETs. Neutrophils are able to produce NETs composed of an exposed DNA scaffold decorated with neutrophil-associated enzymes. A schematic illustration of NETs by neutrophils and the potential targets to block the possible effects of NETs in coculture. Exemplary in vitro NET formation (**right**) by neutrophils cultured in medium (N CM) for 24 h. Cells were stained with citrullinated histone H3 (H3cit) (green), myeloperoxidase (MPO) (red) and Hoechst 33,342 (blue; DNA staining). Cropped versions of the images are provided in (**B**). Scale bar = 5 µm. (**B**) NET formation in cocultures (white arrows). Colocalization of citrullinated histone H3 (H3cit) (green) and myeloperoxidase (MPO) (red) in CM and VC-treated conditions for neutrophils cultured alone or in coculture with MaMel51 cells. Hoechst 33,342 (blue) was used to visualize DNA. Magnified and cropped image is shown on the right for merged image (DNA/H3Cit/MPO). Depicted image area as indicated with numbers. Scale bar = 50 µm. Fluorescence staining of one experiment is shown. (**C**) Dissolving the DNA-scaffold of NETs, using DNase I treatment (5 µg/mL) in VC-treated nonadherent melanoma cell–neutrophils cocultures, did not affect the protective activity of neutrophils. (**D**) Inhibition of proteases by a protease inhibitor mix, from Roche or Selleckchem, for 24 h in cocultures of neutrophils and melanoma cells resulted in blockage of the protective effect toward MaMel51 cells in VC-treated conditions (1 µM vemurafenib and 100 nM cobimetinib). Respective controls with H_2_O (inhibitor (Roche) control) or DMSO (inhibitor (Selleckchem) control) were included for the respective protease inhibitors with equivalent concentrations. The mean MaMel51 cell viability + SD is shown for two to three independent experiments. ns, *p* > 0.05, * *p* ≤ 0.05, ** *p* ≤ 0.01 and *** *p* ≤ 0.001.

**Table 1 cancers-16-01767-t001:** Patient samples and healthy donors used for the characterization of the peripheral blood neutrophil phenotype post-isolation and in the in vitro functional analyses ^1,2^.

Variables			Donor	%
Individual donors			60	100
Patient	Stage	I/II ^3^	3	5.07
		IV ^4^	40	66.6
Healthy donor			17	28.3

^1^ For phenotyping of peripheral blood neutrophils, no clinical parameters were collected. ^2^ Patients and healthy donors overlap with the cohort published by Wendlinger et al. [35]. ^3^ One patient with stage IB, one patient with stage IIA and one patient with stage IIC. ^4^ Stage IV samples were collected from treatment-naïve patients.

**Table 2 cancers-16-01767-t002:** Serum samples from patients diagnosed with advanced melanoma receiving either dual-targeted therapy (total *n* = 50) or immunotherapy (total *n* = 95) were included for biomarker evaluation ^1^.

Variable		Patients	
Age	Median (range)	70 years (36–91)	%
Individual patients		145	100
Sex	male	83	57.2
	female	39	26.9
	unknown ^2^	23	15.9
Stage	III	14	9.7
	IV	131	90.3
M-category	M1a	12	8.3
	M1b	37	25.5
	M1c	67	46.2
	M1d	11	7.6
First-line therapy	yes	134	92.4
	no	11	7.6
Therapy after study inclusion	anti-PD-1	60	41.4
	anti-PD-1 + anti-CTLA-4	32	22.1
	BRAFi + MEKi ^3,4^	48	33.1
	ImmunoCobiVem or others	5	3.4
LDH	>1× ULN ^5^	62	42.8
	<1× ULN	83	57.2

^1^ Patients overlap with cohort published by Wendlinger et al. [35]. ^2^ This information was not provided for serum samples from Tübingen/Erlangen. ^3^ All patients receiving BRAFi + MEKi harbored a BRAFV600-mutation [35]. ^4^ For correlation with galectin-3 levels, values for absolute leukocyte count and absolute and relative neutrophil counts were collected. ^5^ Upper limit of normal (ULN) [35].

**Table 3 cancers-16-01767-t003:** Melanoma and nonmelanoma cancer cell lines tested in this study.

Cell Line	Accession	Disease	Site of Derivation	BRAF	TERT	TP53
451LU; xenograft: WM164	CVCL_6357	cutaneous melanoma	metastatic; arm; skin metastatic; established from lung of a nude mouse.	V600E; heterozygous	n.a.	mut
MaMel51	CVCL_A186	melanoma	metastatic; lymph node	V600E	mut	n.a.
WM3734	CVCL_6800	melanoma	metastatic; brain	V600E; heterozygous	n.a.	n.a.
MaMel86c	CVCL_C7TP	cutaneous nodular melanoma	metastatic; lymph node	V600E	n.a.	mut
Sk-Mel-5	CVCL_0527	cutaneous melanoma	metastatic; axillary lymph node	V600E	mut	wt
MaMel06	CVCL_A119	cutaneous nodular melanoma	metastatic; lymph node	V600E	mut	n.a.
M14	CVCL_1395	amelanotic melanoma	metastatic; right buttock; hypodermis; subcutaneous	V600E; heterozygous	n.a.	mut
MaMel63a	CVCL_A198	melanoma	metastatic; hypodermis; skin/ cutaneous/subcutaneous	V600E	mut	n.a.
Sk-Mel-28	CVCL_0526	cutaneous melanoma	in situ, skin; melanocyte of skin	V600E; homozygous	mut	mut
UACC257	CVCL_1779	melanoma	unknown	V600E; heterozygous	mut	wt
MDA-MB-435 derivative: M14	CVCL_0417	amelanotic melanoma	metastatic; right buttock; hypodermis; subcutaneous	V600E; heterozygous	n.a.	mut
H460	CVCL_0459	lung large cell carcinoma	metastatic; pleural effusion	n.a.	n.a.	wt
A549	CVCL_0023	lung adenocarcinoma	in situ; lung	n.a.	n.a.	wt
WaGa	CVCL_E998	Merkel cell carcinoma, cutaneous neuroendocrine carcinoma	metastatic; ascites	n.a.	n.a.	n.a.

Abbreviations: mut = mutated, wt = wild-type, n.a. = not available.

**Table 4 cancers-16-01767-t004:** Protection of melanoma cells by blood neutrophils after dual BRAF/MEK inhibition or cisplatin treatment.

Treatment with VC		Treatment with CIS	
Cell Lines	Protection in % (Mean of n3–4)	Cell Lines	Protection in % (Mean of n3–4)
451LU	+42.0%	MaMel86c	+37.2%
MaMel51	+34.7%	WM3734	+30.9% (n3)
WM3734	+21.0% (n3)	451LU	+27.9%
MaMel86c	+17.4%	UACC257	+22.9%
Sk-Mel-5	+16.3%	Sk-Mel-5	+19.4%
MaMel06	+14.4%	M14	+19.1%
M14	+13.6%	MaMel51	+17.6%
MaMel63a	+6.9%	MaMel63a	+12.7%
Sk-Mel-28	+2.5%	Sk-Mel-28	+10.0%
UACC257	−2.8%	MDA-MB-435	+6.2%
MDA-MB-435	−2.4%	MaMel06	−5.9%
*H460*	−*8.1% (n3)*	*A549*	*+17.4%*
*A549*	−*11.2%*	*H460*	*+0.9%*
*WaGa*	−*40.6%*	*WaGa*	−*19.7%*

Melanoma and nonmelanoma cancer (*italic*) cell lines are listed with the percentage of protection by neutrophils when exposed to (**left**) 1 µM vemurafenib and 100 nM cobimetinib or to (**right**) 20 µM cisplatin. Viability was normalized to control (T+CM) and protection from cytotoxicity is listed as viability of T:N+VC minus viability of T+VC. Protection is indicated as a positive value. Cytotoxicity by neutrophils toward cell lines are indicated as a negative value. Protection was defined as an increase in viability by at least 10% when cultured with neutrophils.

## Data Availability

The senior author of this study can provide anonymized patient and experimental data upon request for academic studies.

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
