# Peer review of "Susceptibility of Melanoma Cells to Targeted Therapy Correlates with Protection by Blood Neutrophils"

_cancers, 2024, doi:10.3390/cancers16091767_

Round 1

Reviewer 1 Report (Previous Reviewer 2)

Comments and Suggestions for Authors

The manuscript entitled “Susceptibility of Melanoma Cells to Targeted Therapy Correlates with Protection by Blood Neutrophils” revealed protective effects of neutrophils to melanoma cells treated with targeted therapies.  The responses to the initial phase of assessment are notably comprehensive and serve to enhance comprehension in several respects. Nevertheless, there are several remaining points that require attention in order to render the manuscript suitable for publication.

1.       The sections 3.1 and 3.2 on neutrophil characterization (CD16) and Galectin-3 concentration in the patient sera do not represent urgent results for me, and thus I believe that these could be included in the supplement rather than at the beginning of the paper. It is notable that these data are not referenced in the subsequent sections of the work or in the final explanation of the protective effect. Furthermore, Figure 3D demonstrates that the protective effect is independent of whether the neutrophils originate from healthy donors or from melanoma patients.  

2.       In contrast to the previous point, supplemental figure 5 is an important figure in explaining the protective effect of neutrophils and should therefore be included in the main part of the manuscript.

3.       Did the authors observe a protective effect by neutrophils on less susceptible cells, such as SK-Mel-28, when VC concentrations were increased? It can be hypothesized that these melanoma cells would exhibit a stronger response after increasing VC concentrations. If neutrophils are protective, it can be expected that these cells would also benefit from this protective effect.

4.       The data presented in Figure 5 prompts further questions regarding the interpretation of the results. The spheroids demonstrate a clear reduction in size when treated with targeted therapies and co-cultured with neutrophils. This raises the possibility that the observed effect may be the result of the selection of a resistant cell population (as mentioned in the previous review). The vitality graphs shown are always relative figures in percent. However, it is also important to consider the absolute cell numbers per ml of medium. Given the significantly smaller spheroids, it is reasonable to hypothesize that the cell count will also be significantly lower. However, this may contain a higher proportion of vital cells. If this is the case, it is not possible to conclude that the neutrophils are exerting a protective effect, but rather that they are exerting an additional selective pressure, resulting in a small number of vital and robust melanoma cell clones.  

In conclusion, the results are highly descriptive, yet a clear explanation for the mechanisms of the observed effects is still lacking in the current manuscript.

In light of the queries and shortcomings identified, it is recommended that the manuscript undergo a significant overhaul or be rejected outright.

Author Response

Dear Reviewer 1, 

We want to thank you for your critical but very helpful comments and suggestions, which notably refined and improved our manuscript. Enclosed (please see the attachment) we provide our responses and additional information to the raised comments. We hope we could address and clarify each request, making our manuscript suitable for publication. 

Sincerely, 

Simone Wendlinger 

Reviewer 2 Report (Previous Reviewer 3)

Comments and Suggestions for Authors

Authors have satisfactorily answered the comments that were raised during the initial review of this manuscript.

However, I'm not satisfied with the comment#1 for the low sample size and also it has been not discussed in the discussion section as opposed to rebuttal letter.

Need clarification and justification for the low sample size and explicitly mention this in the discussion section indicating this as a limitation of the study.

Author Response

Dear Reviewer 2, 

We want to thank you for your comments and considering our manuscript for publication. We apologize for the missing implementation of the study limitation in terms of low sample size for the ex vivo phenotypic characterization.
The requested information was added to the revised manuscript. Enclosed (please see the attachment) we provide our responses and the information to the raised comment. We hope we could address and clarify the request, making our manuscript suitable for publication. 

Sincerely, 

Simone Wendlinger 

Reviewer 3 Report (Previous Reviewer 1)

Comments and Suggestions for Authors

Dear Authors,

This manuscript is interesting and can be considered for publication.

Comments on the Quality of English Language

Minor English Editing is required.

Author Response

Dear Reviewer 3, 

We want to thank you for your positive feedback and for recommending our manuscript for publication in Cancers. The manuscript underwent refinement and optimization. We assume that minor English language editing will be conducted by the editorial team at Cancers following its acceptance.

Sincerely, 

Simone Wendlinger 

This manuscript is a resubmission of an earlier submission. The following is a list of the peer review reports and author responses from that submission.

Round 1

Reviewer 1 Report

Comments and Suggestions for Authors

This study investigates therapy resistance in melanoma patients, focusing on the role of neutrophils in targeted therapy. Blood neutrophils from advanced melanoma patients showed lower CD16 expression compared to healthy donors. In vitro experiments demonstrated that neutrophils prevented melanoma cell apoptosis under dual MAPK-inhibition, depending on cell-cell contact and melanoma cell susceptibility to treatment, correlated with galectin-3 levels. Inhibition of neutrophil extracellular trap protease activity prevented melanoma cell protection. The study suggests a pro-tumoral neutrophil phenotype in late-stage patients and identifies galectin-3 as a potential marker for melanoma cell protection by neutrophils.

 This manuscript is interesting and impressive, however, but some issues must be solved. I think this manuscript be considered for publication in " Cancers” after revision. My comments are described as following:

 Comments:

1.     Authors used many cell lines in this study, the mycoplasma test must be provided.

2.     The limitation of this study should be addressed.

3.     Authors demonstrated that under MAPK inhibition, neutrophils were assessed for their protection against melanoma cell lines. The expression of MAPK-dependent key molecules must be evaluated.

4.     Galectin-3 appears to be a potential marker for neutrophil-mediated melanoma cell protection. The level of Galectin-3 must be detected by western blotting.

5.     How many cell-culture passages were used in this study?   

6.     The manuscript must consult an English language editor and write in an "academic English" as well as "must" provide proof of a certificate of editing.

Comments on the Quality of English Language

Dear Editor, 

This manuscript can be considered for publication after revision. Many thanks.

Author Response

Dear Reviewer 1, 
we thank you very much for your comments, which we implemented into the revised and restructured manuscript. 
Please find the point-by-point response in the attachment. 
Sincerely, 
Simone Wendlinger

Reviewer 2 Report

Comments and Suggestions for Authors

The manuscript describes the authors' observation that neutrophils protect melanoma cells when treated with targeted therapies. The authors suggest that the protective mechanism is due to the release of neutrophil extracellular traps (NETs). In addition, the authors revealed that the protection of melanoma cells by neutrophils is only observed in non-adherent culture conditions.

The manuscript's structure requires improvement. The high number of citations gives it a review-like quality.   Additionally, the results section contains lengthy introductions at the beginning of each paragraph, sometimes taking up half the paragraph. These introductions should be condensed and integrated into the introduction or discussion sections. This would enable the reader to focus on the crucial data while reading the results. Additionally, it could significantly reduce the length of the manuscript, which is currently quite long.

The headings of the individual paragraphs should summarize the main findings. However, some of the headings in the present manuscript are misleading. For example, the heading 3.1 'Peripheral neutrophil phenotype differs between patients and healthy donors' is imprecise. Although the results indicate that patients with high melanoma stage show a reduction in surface expression of CD16, it is too much to write about a different phenotype, especially since no significant differences were observed in the other markers measured. The heading of section 3.2 "Low levels of galectin-3 and galectin-1 are associated with response to immunotherapy but not to targeted therapy" is also misleading, as the authors did not measure galectin-1 or examine response to immunotherapy.

The submitted manuscript requires significant formula revision and restructuring as currently presented.

The data and results presented are not always related, and the interpretation of the results is not always understandable.

1.)    Figure 1. The results show a decreased CD16 expression on the surface of neutrophiles in melanoma patients with high stage, however the it was not changed in the in vitro coculture experiments. Why are these results important for the manuscript? The neutrophils used in the in vitro experiments were obtained from healthy donors. Since most of the results show data from short co-cultures (24 hours), there is some doubt that the normal neutrophils change their phenotype in a tumorigenic state.   

2.)    Figure 2. Is there an explanation for the change of patient numbers for the different analysed parameters? (e.g. Galectin – 3 and MPO pre treated)

3.)    The results of figure 2 and 3 are inconsistent. While in figure 2 no differences of galectin-3 was observed between the responders and non-responders, the authors show an increase galectin-3 secretion of cells responding to targeted therapy.  

4.)    Figure 3. Why the author did not use BRAF wild type melanoma cells or cells of other cancer entities with BRAF mutation as control in figure 3.E/F? This would make more sense.

5.)    The authors analysed the viability of tumor cells by a flow cytometry based apoptosis assay. It is necessary to include the flow cytometry plots to be able to better assess the results.

6.)    Is the increased galnectin-3 a result of secretion or cell destruction?

7.)     Figure 4: Shows increased viability of melanoma cells after treatment with targeted therapies when neutrophils are added to the cell culture. This effect appears to be dependent on the melanoma cell:neutrophil ratio. Is it possible that the effect is an artifact due to a different total number of cells in the culture? This would also involve altered culture conditions such as pH, metabolites, etc. This could also affect the uptake or efficacy of the therapy in the melanoma cells. In most experiments, melanoma cells exhibit significant apoptosis after 24 hours of treatment, with only approximately 20% of cells remaining viable. Figure 4E shows a higher percentage of living cells after 48 hours (approximately 35%), despite the addition of neutrophils only occurring after 24 hours. The observation in the apoptosis assay do not necessarily indicate that neutrophils protect melanoma cells. Rather, it is possible that the neutrophils became CFSE-positive due to phagocytosis of CFSE-stained melanoma cells and were mistakenly evaluated as vital melanoma cells. 

8.)    Figure 5. The labels in figure 5D are misleading. The spheroid assays show a clear decrease in spheroid size during treatment and in particular after addition of neutrophils. However, the vitality of the cells seems to increase this is not consistent. Here is again the question if neutrophils were inadvertently measured as vital melanoma cells? Could the targeted therapies and neutrophils be triggering strong selective pressure, resulting in the enrichment of a highly robust and viable melanoma cell population?

9.)    Figure 6.The hypothesis that NETs affect the protective mechanism to targeted therapy treated melanoma cells is not clearly proven. In supplemental figure S6B the authors show induction of NETs in all conditions (with and without tumor cells and/or treatment). There seems to be no difference. In addition, it is noteworthy that DNase did not hinder the protective effect of neutrophils, despite its role in preventing NETs. This is particularly surprising given that the proteases are bound to the DNA, which would be even more crucial in non-adherent cultures to anchor them in place. Here I would expect similar results as shown after protease inhibition.

In conclusion, the results are insufficient to support the observations in a clear and comprehensible manner. The techniques and experiments employed may not be optimal for investigating the mechanisms or interpreting the results accurately. In addition, as the results only apply in non-adherent culture conditions, there should also be a stronger focus on circulating tumour cells (CTCs) in the manuscript.

Therefore, I recommend rejecting the manuscript in its current form.

Author Response

Dear Reviewer 2,
we thank you very much for your critical and highly relevant comments. We adapted our manuscript accordingly. 
Please find the point-by-point response in the attachment.
Sincerely,
Simone Wendlinger

Reviewer 3 Report

Comments and Suggestions for Authors

The study by Wendlinger and colleagues discusses the protective role of circulatory neutrophils against the susceptibility of melanomas towards the immunotherapy. The manuscript is potentially interesting and gives several insights to the field of NETosis and its interconnection with cancer progression. Abstract is clear and concise; titles are informative and reflects the description. Experiments are well designed, executed and the conclusions drawn are supported by the results and are justified. I have few concerns that are summarized below before publication is possible.

1.     What is the reason for low sample size in the cohort 1 and how authors could draw the conclusion form just 3 samples (stage I/II). Needs clarification in the text and justification for inclusion of low samples in the study.

2.     Study cohort 2 looks great. Does these patients have any underlying co-morbidities or secondary infections that could prime the neutrophils for NETs formation.

3.     Are the neutrophils presented in the manuscript is from cohort 1 or 2. Mention this in the methods section.

4.     Be consistent in the usage of vemu and cobi throughout the manuscript including figures.

5.     In fig 6A what stimuli is used for NETs release.

6.     Regarding NETs release authors should provide more data related to this section. Provide with either confocal/fluorescence microscopy images to show the localization of these NETs markers. Just curious to know are these peripheral blood neutrophils from patients were prone to NETs formation?

Author Response

Dear Reviewer 3,
we thank you very much for your comments. We adapted our revised and restructured manuscript accordingly.
Please find the point-by-point response in the attachment.
Sincerely,
Simone Wendlinger
